# Modeling the health and economic implications of adopting a 1-dose 9-valent human papillomavirus vaccination program in adolescents in low/middle-income countries: An analysis of Indonesia

**Vincent Daniels[1]\*, Kunal Saxena[1], Oscar Patterson-Lomba[2], Andres Gomez-Lievano[2], Jarir At Thobari[3], Nancy Durand[4], Evan Myers[5]**

1 Merck & Co, Rahway, NJ, United States of America, 2 Analysis Group, Inc., Boston, MA, United States of America, 3 Universitas Gadjah Mada, Yogyakarta, Indonesia, 4 Temerty Faculty of Medicine, University of Toronto, Toronto, ON, Canada, 5 Department of Obstetrics and Gynecology, Duke University Medical Center, Durham, NC, United States of America

\* vincent_daniels@merck.com

## Abstract

### Background

Recent evidence suggests that 1 dose of the human papillomavirus (HPV) vaccine may have similar effectiveness in reducing HPV infection risk compared to 2 or 3 doses.

### Objective

To evaluate the public health impact and cost-effectiveness of implementing a 1-dose or a 2-dose program of the 9-valent HPV vaccine in a low- and middle-income country (LMIC).

### Methods

We adapted a dynamic transmission model to the Indonesia setting, and conducted a probabilistic sensitivity analysis using distributions reflecting the uncertainty in levels and durability of protection of a 1-dose that were estimated under a Bayesian framework incorporating 3-year vaccine efficacy data from the KEN SHE trial (base-case) and 10 year effectiveness data from the India IARC study (alternative analysis). Scenarios included different coverage levels targeted at girls-only, or girls and boys. Costs and benefits were computed over 100 years from a national single-payer perspective.

### Results

Depending on the coverage and target population, the median number of cancer cases avoided in 2-dose programs ranged between 600,000–2,100,000, compared to 200,000–600,000 in 1-dose programs. The 1-dose programs are unlikely to be cost-effective compared to 2-dose programs even at low willingness-to-pay (WTP) thresholds. The girls-only 2-dose program tends to be cost-effective at lower WTP thresholds, particularly in scenarios

**Data Availability Statement:** The data that support the findings of this study are available in the Appendix file.

**Funding:** This study was funded by Merck & Co., Inc. The study sponsor was involved in all aspects of the research, including the study design, analysis, interpretation of data, and writing of the manuscript.

**Competing interests:** I have read the journal's policy and the authors of this manuscript have the following competing interests: Vincent Daniels and Kunal Saxena are employees of Merck Sharp & Dohme LLC, a subsidiary of Merck & Co., Inc., Rahway, NJ, USA and shareholders in Merck & Co., Inc., Rahway, NJ, USA. Oscar Patterson-Lomba and Andres Gomez-Lievano are employees of Analysis Group, Inc., a consulting company that has provided paid consulting services to Merck & Co., Inc, which funded the development and conduct of this study and manuscript. Jarir At Thobari, Nancy Durand, and Evan Myers have received consultancy fees from Merck & Co., Inc.

with high coverage, dose price and discount rate, while the girls and boys 2-dose program is cost-effective at higher WTP thresholds. In the alternative analysis, 1-dose programs have higher probability of being cost-effective compared to the base-case, particularly for low WTP thresholds (less than 0.5 GDP) and for high coverage, dose price and discount rate.

## Conclusion

Adoption of 1-dose programs with 9-valent vaccine in an LMIC resulted in more vaccine-preventable HPV-related cancer cases than 2-dose programs. The 2-dose programs were more likely to be cost-effective than 1-dose programs for a wide range of WTP thresholds and scenarios.

## Introduction

HPV is a common sexually transmitted infection, and it is the primary cause of cervical cancer and a significant proportion of vulvar, vaginal, anal, penile, and oropharyngeal cancers, genital warts, and recurrent respiratory papillomatosis [1–3]. Thus, HPV is associated with substantial clinical and economic burdens, accounting for an estimated 5% of cancers worldwide [4, 5].

HPV vaccines with 2- or 3-dose regimen have shown to be highly effective in preventing HPV-related disease and cancers, particularly if administered before sexual debut [6, 7]. Currently available vaccines include three 2-valent vaccines targeting HPV types 16/18 (Cervarix, Glaxo Smith Kline; Cecolin, Innovax; Wozehui, Walvax), two 4-valent vaccines targeting HPV types 6/11/16/18 (Gardasil, Merck & Co. Inc; Cervavax, Serum Institute of India), and a 9-valent vaccine targeting HPV types 6/11/16/18/31/33/45/52/58 (Gardasil 9, Merck & Co. Inc.). In 2020, the World Health Organization (WHO) launched the Cervical Cancer Elimination Initiative [8], calling countries to lower the yearly incidence of HPV to 4 cases per 100,000 by 2030 [9]. These disease elimination goals are ambitious [10] (see, e.g., Brisson et al. [2020] [11] for an evaluation of the feasibility and timing of achieving these goals), particularly for low- and middle-income countries (LMICs) where over 80% of cases of cervical cancer cases and more than 90% of cervical cancer deaths occur due to limited resources to implement national prevention, screening and precancer treatment programs to reduce cervical cancer incidence and mortality rates [12, 13].

The 9-valent HPV (9vHPV) vaccine is indicated for administration as a 3-dose regimen. An alternative 2-dose regimen (with a 6- to 12-month interval between doses [14, 15]) is licensed in many countries for use in boys and girls aged 9–14 years [16]. However, in recent years, a growing body of evidence from observational studies [17–19] and real-world settings [20–30] has indicated that 1-dose of HPV vaccine may be sufficient to elicit a protective immune response against incident and persistent HPV infection. Results from the first randomized controlled trial of single-dose efficacy, KEN SHE (NCT03675256) [31], in Kenya indicated that the 9vHPV vaccine efficacy at 36 months was 98.8% (91.3–99.8%) against persistent infection with HPV 16/18 and 95.5% (89.0–98.2%) against persistent infection with HPV 16/18/31/33/45/52/58 [32]. The Dose Reduction Immunobridging and Safety Study (DoRIS, NCT02834637), a randomized trial among Tanzanian girls, showed stable HPV 16/18 antibody levels 24 months after 1-dose 9vHPV or bivalent HPV (2vHPV) vaccine, and no significant differences in antibody avidity between dosing groups (1, 2 or 3 doses) [33, 34]. Results from IVIHPV1 (NCT03747770) also showed that 1-dose 9vHPV elicited persistent HPV16/18 antibody responses [35, 36]. A 10-year follow-up analysis of the India IARC study

(NCT00923702), which has "default" single-dose population, indicated the 2vHPV single-dose efficacy against persistent cervical infection at a mean follow-up of 9 years is 95.4% (95% CI 85.0–99.9) [36]. Taking this evidence into consideration, in December 2022, WHO updated its position paper to provide permissive recommendation of 1-dose HPV vaccination for girls and boys ages 9 through 20 [37]. More evidence on 1-dose HPV vaccination will become available in the coming years from ongoing clinical trials that were designed to study the efficacy of the single-dose schedule, including long-term results from the KEN SHE trial, two ongoing randomized, controlled trials, ESCUDDO (NCT03675256) and PRISMA (NCT05237947), and two recently announced randomized, controlled trials (one in females ages 16–26 years old, and another in males ages 16–26 years old) [38].

If the existing and upcoming evidence definitively indicate that a single dose of HPV vaccine is indeed highly efficacious and durable, it could have significant benefits, such as simplification of vaccine delivery and lowering of program costs, which could in turn accelerate the introduction of HPV vaccines more broadly into worldwide immunization programs in males and females, multi-age cohorts, and adults. These advantages would be most beneficial to LMICs, where the burden of cervical cancer is high yet resources for public health initiatives, treatments, and data systems for monitoring are more likely to be constrained.

Considering the limited randomized controlled trial (RCT) data available and the increasing interest by public health authorities in LMIC setting in adopting single dose programs, it is important to assess the impact of the uncertainty on the effectiveness of a single HPV vaccine dose, especially around its durability, in the evaluation of HPV vaccine programs in the context of LMICs. Dynamic transmission modeling has been used to analyze and compare health and economic outcomes of a 1-dose vs no-dose, or 1-dose vs 2-dose programs [39–41]. The health and economic outcomes of the different vaccination programs will depend on the effectiveness of a single dose, as well as the coverage, coupled with the epidemiologic, demographic, and economic characteristics of the country/setting of interest.

In general, previous model-based studies have found that a 1-dose program is cost-effective compared with no vaccine, and that a 1-dose HPV vaccination regimen could be a viable option in most settings given particular assumptions about 1-dose effectiveness [39–47]. For example, Burger et al. (2018) [39] used a one-way analysis with 10, 15, and 20-year one dose durations and 80% efficacy. Kim et al. (2021) [43] also assumed 80% efficacy against infection and used a few simplified vaccine scenarios, with a minimum of 10 years of full protection followed by linear waning. Meanwhile, Tan et al. (2018) [45] reported the reduced incidence corresponding to a single dose with an assumption of 80% lifelong efficacy. We briefly review these studies in Appendix A in S1 Appendix.

Previous models assessing 1-dose compared to 2-dose vaccine regimen in LMIC settings [39–47] were parametrized with point value assumptions for the 1-dose vaccine parameters and some assumed high vaccination coverage rates in settings that are unlikely to achieve such levels of coverage. These models did not account for the impact of the uncertainty on these 1-dose vaccine parameter values and on the vaccination coverage rates on the model's predicted health and economic outcomes. Given the uncertainty regarding the efficacy and durability of a single dose of HPV vaccine, models should explicitly account for, and characterize the impact of such uncertainty on health and economic outcomes, rather than only exploring the effects of single parameter values in isolation and without considering their relative likelihood (see Daniels et al. [2022] [48]).

The objective of this study is to investigate the health and economic implications of introducing a 1-dose program with 9vHPV vaccine, instead of a 2-dose program, in an archetypical LMIC without an existing vaccination program or cervical screening program (i.e., Indonesia) using a dynamic transmission model (DTM) coupled with a probabilistic sensitivity analysis

(PSA) to estimate the impact of the uncertainty around the 1-dose effectiveness and durability on the health and economic outcomes that are relevant to decision makers.

## Methods

### Model overview

A compartmental, age-structured, deterministic DTM of 9vHPV vaccination [48] was adapted for Indonesia. The model included all major HPV disease-related endpoints (i.e., cervical, vaginal, and vulvar cancers and pre-cancers, anal, penile, oropharyngeal cancers, genital warts, and recurrent respiratory papillomatosis [RRP]), and key disease transitions, such as HPV acquisition, transmission, recovery, re-infection, natural immunity, progression to pre-cancerous lesions, development of cancer, disease detection, treatment, regression, and vaccination.

The model assumed a 'leaky' vaccine where all uninfected vaccinees have partial protection ('degree of protection') that may wane. To better fit the existing data on single-dose efficacy and durability of protection (in which vaccine protection is relatively sustained initially followed by a steep decay), the model was modified so that the mean duration of vaccine protection was modelled as a Gamma distribution, whereas earlier versions assumed a constant waning rate resulting in an exponential decay of protection. This waning model is also consistent with the waning profile used in other single-dose modeling studies, such as the HPV-ADVISE model (which used a normally distributed duration of protection) [49] among others [11, 40, 41, 45]. The mean duration of protection is defined as the inverse of the waning rate parameter in the model. The new waning model and parameter estimations are described and discussed in detail in Appendix B.2 in S1 Appendix. Note that "efficacy" in the presence of waning is not equivalent to the model parameter "degree of protection", but it is a function of both degree of protection and the waning rate (see Equation 2 in Appendix B.2 in S1 Appendix).

The complete-series (2-dose) vaccine degree of protection was assumed equal to the vaccine's prophylactic efficacy from clinical trial data [6, 50–54], and the duration of protection against HPV genotypes 6/11/16/18/31/33/45/52/58 was assumed to be lifelong for complete-series vaccination (i.e., 2-dose regimen administered at age 13 years). See Appendix C section 4 in S1 Appendix for full series parameter values. Details and validation of the model have been previously published [48, 55–57].

**Demographic and economic assumptions and other model inputs.** The model assumed a constant population size of 273,523,621 persons (137,717,861 male and 135,805,760 female) and a constant age distribution throughout the simulations [58]. Sexual mixing details in the population model structure are described in previous publications [48]. The population is divided into low, medium, and high sexual activity groups with an average annual number of new sexual partners for each activity level. The number of new sexual partners by age group was estimated from lifetime partnership data from Indonesia and a model distributing these new partnerships over ages (described in Appendix D in S1 Appendix). The model was calibrated to the Indonesia epidemiological incidence data for cervical intraepithelial neoplasia (CIN), cervical, vaginal, vulvar, anal, oropharyngeal, and penile cancers and genital warts (calibration targets and details in Appendix E in S1 Appendix). The model assumed no historical vaccine coverage in this population (i.e., no existing national immunization program for HPV as of the start of the simulations), and that no cervical screening program was in place. Natural history and other remaining inputs are in Appendix C in S1 Appendix or previous model publications [48, 57, 59].

Costs and benefits were computed from the perspective of a national single payer over 100 years beginning in 2022 with a 3.0% annual discount rate for costs and quality-adjusted life

years (QALYs). Cost and other economic inputs are detailed in Appendix F in S1 Appendix. Briefly, vaccine prices were derived from the Setiawan et al. CE analysis for Indonesia [60] and were in line with WHO-provided vaccine purchase data for GAVI-supported countries [61]. We assumed a vaccine price of $20 United States dollars (USD) per dose including delivery and administration costs that is intended to represent a potential GAVI price. The per-dose price and administration costs were assumed to be the same for a 1-dose program or 2-dose standard of care program.

**Model outputs.**   Model outputs covered epidemiological outcomes, including total and incremental cases (comparing each vaccination strategy to no vaccine) of HPV-related cervical, vaginal, vulvar, anal, oropharyngeal, penile cancers, genital warts and RRP. The model also estimated discounted costs and QALYs, and difference in net monetary benefit (NMB) to generate cost-effectiveness acceptability curves [48]. All cumulative health and economic results are calculated beginning in 2022. A PSA was conducted to assess the impact on model outcomes of the uncertainty around the degree and duration of protection model parameters of a single dose of HPV vaccine (see details below). Finally, we assessed time to HPV-related cervical cancer disease elimination, defined as either <10 cervical cancer cases per 100,000 or <4 cervical cancer cases per 100,000.

**Distribution of degree and duration of protection from single dose of HPV vaccine.** To rigorously model the health and economic impact of a single dose HPV vaccination program in a LMIC, we first need to ascertain the values characterizing the 1-dose vaccine model parameters, namely degree and duration of protection.

Based on the empirical comparative evidence for the efficacy and effectiveness of a single dose [20, 21], it was assumed that the degree and duration of protection of a single dose of HPV vaccine was not superior to that of two doses. The degree of protection of a single dose was modeled as a fraction of the protection provided by two doses. It was also assumed that the duration of protection and the fractional reduction in degree of protection of a single dose was the same for all vaccination efficacy endpoints (i.e., the protection against transient and persistent infection, as well as disease endpoints) but different for different HPV vaccine types based on available data.

Efficacy data from the KEN SHE trial in Kenya for incidence of persistent HPV 16/18 and HPV 16/18/31/33/45/52/58 pooled populations at 18 and 36 months offers empirical evidence for estimating the joint probability distribution for the degree and duration of protection model parameters from a single dose of HPV vaccine. Herein, these data were used to estimate the joint probability distribution based on a Bayesian inference approach. The KEN SHE trial only measured the outcome of persistent cervical infection. Specifically, the data from the pooled HPV 16/18 population was used to generate parameter distributions for HPV 6, 11, 16, 18, and the pooled HPV16/18/31/33/45/52/58 data was used to generate the distributions for HPV 31/33/45/52/58, covering all nine vaccine types. We also estimated an alternative distribution using the same KEN SHE data in combination with efficacy data from the India IARC 10-year follow-up study results [36]. These India IARC efficacy estimates were applied to all vaccine HPV types equally. This was considered the alternative distribution because the vaccine effectiveness evidence from India IARC is arguably less robust than the efficacy data from KEN SHE due to limitations of the IARC study (e.g., non-randomized design which can lead to confounding and bias [62]). Moreover, using the KEN SHE data, in combination with the IARC data, resulted in higher estimates for 1-dose durability of protection in the short-term (during the first 10 years) compared to an estimation approach using only the IARC data. A detailed description of these trial data and the methods used to generate distributions can be found in Appendix B in S1 Appendix.

For the parameter estimation it was assumed that the degree of protection for a single dose was no greater than the 2-dose degree of protection against persistent cervical infection (which is estimated to be 0.988 [52]), and that the duration of protection was at least 36 months, which was the latest available time point from the KEN SHE trial. The model-fitted degree of protection distribution for a single dose was then scaled by 0.988 to represent a degree reduction factor. This resulted in an empirical joint probability density distribution of the degree reduction factors and duration of protection for persistent cervical infection, which we then applied to the single dose vaccine model properties for both sexes and all disease endpoints. See more details in Appendix B in S1 Appendix.

The joint posterior distribution for the reduction factor of the degree and the duration of protection parameters was then used for the PSA. Specifically, we randomly sampled 500 sets of points from the joint posterior distributions, and ran the DTM using different values for the degree and the duration of protection from a single dose HPV vaccine to characterize the uncertainty around the corresponding health outcomes and cost-effectiveness. We also performed a similar sampling of the alternative distributions.

### Vaccination scenarios

Two populations eligible for vaccination were modeled:

1. Girls-only vaccination, 9-14-year-olds

2. Girls and boys or gender-neutral vaccination, 9-14-year-olds (GNV)

   Three vaccination programs were considered (with 9vHPV vaccine):

1. The *standard of care*: 2-dose program where a certain percentage initiate the vaccination series (i.e., receive one or more doses), and a percentage of these complete the series (see coverage scenarios below)

2. Hypothetical 1-dose: 1-dose program with the same vaccination initiation rate (i.e., percentage who receive a single dose of the HPV vaccine) as the standard of care scenario

3. Status quo: 'no-vaccination'

The combination of the two eligible populations (girls-only or GNV) for vaccination with the 2-dose and 1-dose programs determine four vaccination strategies. These four strategies are then considered under the three coverage scenarios in Table 1. Note that in the 2-dose low coverage scenario, more individuals receive one dose than two doses due to completion rate being less than 50%.

The initiation and completion rates scenarios were based on the Bruni et al. (2020) [10] estimates of national HPV immunization coverage. The data (Fig 5 of Bruni et al. [2020]) were grouped into three groups of high 1+ dose coverage (66% - 100%), medium (33% - 65%), and low (0% - 32%). Then we calculated the average 1+, average 2-dose, and the average compliance (average 2-dose / average 1+ dose) to arrive at three coverage scenarios used above. Considering the data indicate that $2^{nd}$ dose completion is positively correlated with 1+ dose coverage, we did not examine combinations such as medium coverage with high series completion.

The vaccination scenarios were evaluated using 1-dose parameters drawn at random from the joint posterior distributions of degree and duration of protection factor. In all scenarios, 2 doses were assumed to provide permanent protection and degree of protection as described in Daniels et al. (2022) [48]. Relevant health and economic outcomes were calculated for each of the 500 pairs of the 1-dose vaccine regimen properties to generate distributions for each outcome.

**Table 1. Vaccine coverage scenario for the 1-dose and 2-dose programs.**

| Vaccine program | High-Level Coverage | Mid-Level Coverage | Low-Level Coverage |
|---|---|---|---|
| *2-dose program* | *89% receive one or more doses, 89% of those complete 2 dose series* | *76% receives one or more doses, and 67% of those complete 2-dose series* | *35% receives one or more doses, and 41% of those complete 2-dose series* |
| % with 2-doses | 79.2% | 50.9% | 14.4% |
| % with 1-dose | 9.8% | 25.1% | 20.6% |
| % with 0 dose | 11% | 24% | 65% |
| *1-dose program* | *89% receives one dose* | *76% receives one dose* | *35% receives one dose* |
| % with 2-doses | 0% | 0% | 0% |
| % with 1-dose | 89% | 76% | 35% |
| % with 0 dose | 11% | 24% | 65% |

## Cost, QALY, and NMB calculations

Cost and QALYs for the 1-dose and standard of care scenario HPV vaccination programs were calculated according to the methodologies described in Daniels et al. (2022) [48] and Daniels et al. (2021) [57]. Cost-effectiveness assessments were based on cost-effectiveness acceptability curves and willingness-to-pay (WTP) thresholds of 0.5 and 1, 2, and 3 times the gross domestic product (GDP). In addition, we consider the cost-effectiveness frontier for expected values of total costs and QALYs, which is determined by first finding the mean values of the cost and QALY for each of the sampled vaccine parameter pairs, and then determining the cost-effectiveness frontier and incremental cost-effectiveness ratios (ICERs) for those values. Costs and QALYs were each discounted by 3% in the base case and accumulated over a 100-year time horizon.

## Sensitivity analyses

One-way sensitivity analyses were conducted to consider changes in the assumed coverage of one or more doses (high, medium, and low coverage), per dose cost ($10 or $60 per dose), the assumed discount rate (0% or 5%) for costs and QALYs, costs (base case costs +/- 20% for all costs), and health state utilities (base cases +/- 20%). We also conducted the same sensitivity analyses using the alternative distributions described above.

## Results

### Distributions of degree and duration of protection for a single dose

The base-case distribution of the vaccine model parameters (i.e., degree and duration of protection for a single dose) fitted to the KEN SHE data (pooled HPV 16/18/31/33/45/52/58 and pooled HPV 16/18 study populations) at 18 and 36 months, and the alternative distribution from fitting the same KEN SHE data in combination with the India IARC 10-year follow-up data, are presented in Appendix B in S1 Appendix. For the base-case analysis (fitting to the KEN SHE data alone), the median and 95% confidence interval (CI) of the degree factor are 98.4% [94.7, 99.9%] for HPV 6/11/16/18, and 94.4% [89.2%, 98.0%] for HPV 31/33/45/52/58, which are consistent with the vaccine efficacy against persistent HPV 16/18 infection (98.8%)

**Table 2. Probability of 1-dose vaccine parameter properties achieving certain values.**

| Distribution | HPV Types | Probability degree of protection factor ($f_\psi$) is | | | | Probability duration of protection ($t_\mu$) is | | | | |
|---|---|---|---|---|---|---|---|---|---|---|
| | | >0.80 | >0.90 | >0.95 | >0.99 | >10y | >20y | >30y | >50y | >100y |
| Base | 6,11,16,18 | 100% | 100% | 96% | 31% | 43% | 20% | 12% | 6% | 3% |
| | 31,33,45,52,58 | 100% | 95% | 39% | 0% | 39% | 18% | 10% | 5% | 2% |
| Alternative[1] | 6,11,16,18 | 100% | 100% | 98% | 32% | 97% | 49% | 31% | 16% | 7% |
| | 31,33,45,52,58 | 100% | 100% | 55% | 0% | 99% | 61% | 38% | 20% | 9% |

1 The results for the alternative distribution show a counter-intuitive pattern between the distribution of degree of protection and duration of protection. In general, it is expected that if an HPV type group has higher probability of protection compared to another group, it should also have higher probability of duration. This is the case for the base distribution, but not so for the alternative distribution. Specifically, note how the protection numbers are higher for the 6/11/16/18 group, but the duration numbers are lower for that group. The reason behind this, is that, on one hand, for the 6/11/16/18, the efficacy estimates from KEN SHE (97.5 at 1.5 and 98.8 at 3 years) are higher than the effectiveness of 95.4% at 9 years from IARC, which leads to shorter durations being more likely. On the other hand, for the 31/33/45/52/58 all-type case, the efficacy estimates from KEN SHE (88.9 at 1.5 and 95.5 at 3 years) are lower or similar than the same effectiveness of 95.4% at 9 years from IARC, which leads to a lower degree of protection but also to a longer duration. This can be observed visually in Fig 7 in Appendix B in S1 Appendix.

and against HPV 16/18/31/33/45/52/58 infection (95.5%) estimated at 36 months in the KEN SHE study. The corresponding median and 95% CI for duration of protection are 9.1y [4.3y, 121.9y] and 8.3y [3.8y, 98.7y]. For the alternative distributions (fitting to the KEN SHE and the IARC data), the median and 95% CI of the degree factor are 98.4% [95.1%, 99.9%] for HPV 6/11/16/18 and 95.2% [91.4%, 98.1%] for HPV 31/33/45/52/58. The corresponding median and 95% CI for duration of protection are 20.3y [10.3y, 301.5y] and 24.6y [11.6y, 333.3y]. Thus, as expected, including the India IARC data results in a longer duration of protection with larger CI, although the degree-of-protection factors are similar, but with a narrower CI. To help further characterize these posterior distributions, Table 2 shows various probabilities for each of the model parameters given by the fitted distributions. For example, the probability of duration of protection being greater than 20 years is about 18–20% and 49–61% for the base and alternative distributions, respectively, while the probability of the degree of protection factor being greater than 0.95 is about 39–96% and 55–98% for the base and alternative distributions, respectively.

Table 3 shows the joint probabilities of achieving certain ranges of pairs of parameter values that have been used in previous models, specifically, the probability of having "perfect" protection with more than 20 years duration of protection or having 80% protection for "lifetime". The estimates indicate that none of these parameter combinations are very probable, even when considering the IARC long-term effectiveness data. For example, for prevention of persistent HPV infection, the probability of having duration more than 20 years and degree of protection >0.99 ("perfect protection") ranges from 0.04 to 6.12% (depending on HPV type) or 0.2 to 15.3%, for the base-case and alternative distributions, respectively. Similarly, the probability of a degree of protection greater than 80% with more than 60 years ("lifetime") protection ranges from 5.6 to 4.62% (depending on HPV type) or 13.3% to 16.70%, for the base-case and alternative distributions, respectively.

**Table 3. Joint probabilities of model parameters meeting certain value ranges.**

| Distribution | HPV Types | $f_\psi$ >0.99 & $t_\mu$>20 ("perfect" protection for at least 20 years) | $f_\psi$ >0.80 & $t_\mu$>60 (80% "lifetime" protection) |
|---|---|---|---|
| Base | 6,11,16, or 18 | 6.12% | 5.06% |
| | 31,33,45,52, 58 | 0.04% | 4.62% |
| Alternative | 6,11,16, 18 | 15.3% | 13.30% |
| | 31,33,45,52, 58 | 0.20% | 16.70% |

## Health outcomes

Fig 1 shows the model-estimated cumulative number of HPV-related cancer cases avoided over a 100-year time horizon for each vaccination strategy (blue and purple boxes represent the GNV and girls-only 2-dose strategies, respectively, while orange and yellow boxes represent the corresponding single-dose strategies), as compared to no vaccination. The results are presented for all cancers and by cervical and non-cervical cancers. Each coverage scenario is shown in three separate column panels, while the results for the base-case and alternative distributions are presented in the top panel A and lower panel B, respectively. The ranges in the boxplot reflect the impact on outcomes of the variability of the joint posterior distributions of the degree factor and duration of protection from a single-dose regimen.

For the base-case distribution (panel A), the results indicate that most cancers avoided with any vaccine program are cervical cancers. Also, the GNV programs avoid more cancer cases, on average, than the corresponding girls-only programs, but are not significantly differentiated. Moreover, for all levels of coverage, the 2-dose programs avoid more cancer cases, on average, than the 1-dose programs, with these differences being more pronounced as coverage increases. In fact, higher coverage/completion is associated with larger number of cases avoided (as expected), but also with greater differentiation between the 1- and 2-dose vaccine programs. As a result, the high coverage/completion rate scenario (right-most panels) avoids most cancer cases and features a strong differentiation between the 1- and 2-dose programs in both magnitude and uncertainty. For the alternative distributions (panel B), results indicate that, compared to the base-case results, the number of cases avoided based on the alternative distributions is higher for each dosing strategy, which is expected given the longer duration of protection for the 1-dose schedule in the alternative scenario. Moreover, using the alternative distributions, the median number of cases avoided by 1-dose moves closer to the 2-dose result for any given strategy and coverage. The range of variation in outcomes given the alternative distribution is about 25% smaller than that of the base-case distribution almost entirely provided by increased lower bound. Here too, the GNV programs lead, on average, to more cancer cases avoided than the corresponding girls-only programs.

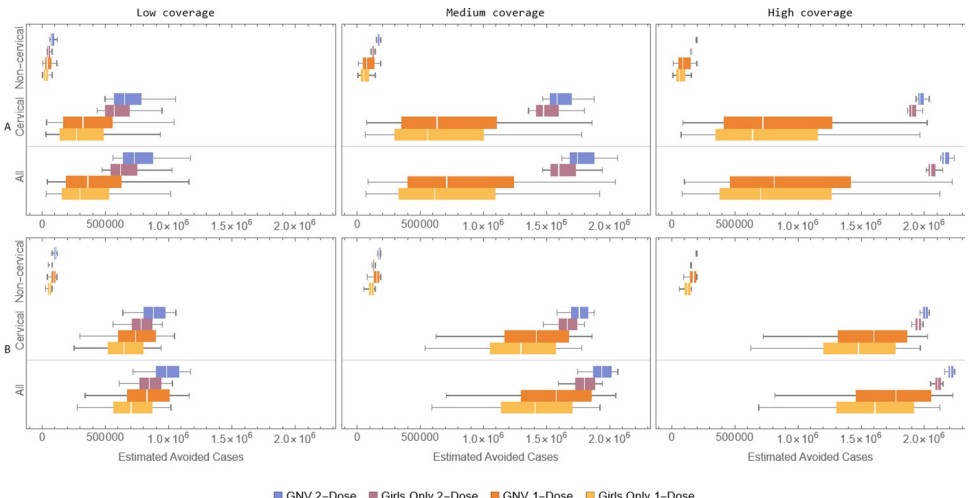

**Fig 1.** Estimated cancer cases avoided by coverage scenario for the four different vaccination strategies compared to no vaccination, over a 100-year horizon for the base-case (A) and alternative (B) distributions. The figure shows all cancers combined and the portions due to cervical and non-cervical cancers. The boxes in the boxplots cover the 25% to 75% quartiles, the white vertical lines represent the median and whiskers represent the minimum and maximum values.

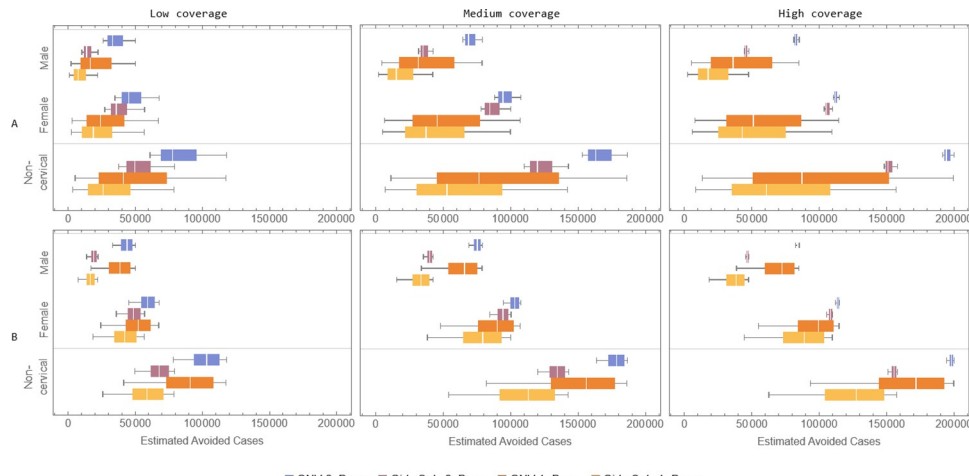

**Fig 2.** Estimated non-cervical cancer cases avoided compared to no vaccination by coverage scenario for the four different vaccination strategies, over a 100-year horizon for the base-case (A) and alternative (B) distributions. The figure shows all non-cervical cancers combined and the portions due to female and male cancers. The boxes in the boxplots cover the 25% to 75% quartiles, the white vertical lines represent the median and whiskers represent the minimum and maximum values.

Fig 2 shows model-estimated cumulative number of HPV-related non-cervical cancer cases avoided over a 100-year time horizon for each vaccination strategy as compared to no vaccination for the base-case (panel A) and alternative (panel B) distributions. The results are presented for all non-cervical cancers and divided into male and female cancers. In the base-case, male cancers represent a substantial portion of the non-cervical cancers avoided. Moreover, the 2-dose programs tend to avoid, on average, more non-cervical cancer cases than the 1-dose programs, and the GNV program (1 or 2-dose) always avoids more male cancers than the corresponding girls-only program (due to direct protection against males and their cancers). Thus, the GNV 1-dose program can have similar or better impact compared to the girls-only 2-dose program for avoiding male cancers, particularly in the low and medium coverage scenarios, albeit with more uncertainty. The results for the alternative distributions are similar, except that now the GNV 1-dose program avoids more male cancer cases, on average, than the girls-only 2-dose program, but with more uncertainty.

## Cost-effectiveness

Fig 3 shows the cost-effectiveness acceptability curves, for each of the five different vaccination strategies (including "no vaccine") for the base-case (panel A) and alternative (panel B) distributions, assuming a per dose cost of $20 USD. The curves show the probability of each strategy being cost-effective as a function of WTP. The sum of all probabilities for all five strategies at a given WTP is 100%.

For the base-case distribution (panel A), in the low and medium coverage scenarios, the 2-dose GNV (purple curve) has nearly 100% probability of being cost-effective at the 1/2 GDP per QALY WTP threshold and above, while in the high coverage scenario, the girls-only 2-dose is cost-effective at the 1/2 GDP per QALY threshold and below. As the coverage increases, so does the range of WTP values for which the girls-only 2-dose program has a high probability of being cost-effective. Neither of the 1-dose programs show a substantial probability of being cost-effective even at the lowest WTP thresholds. For the alternative distributions (panel B), the probability of 1-dose programs being cost-effective is higher in terms of

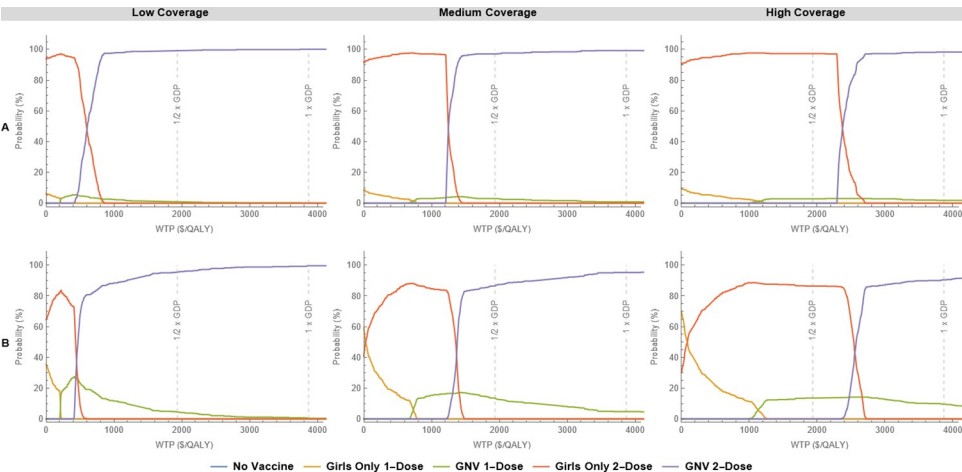

**Fig 3.** Cost-effectiveness acceptability curves by coverage scenario, for each of the four different vaccination strategies, assuming $20 USD per dose, for the (A) base-case and (B) alternative distributions. The vertical dashed lines represent the WTP thresholds for 0.5, 1 and 2 times GDP per QALY, respectively.

acceptability curves, particularly the girls-only 1-dose for low WTP thresholds and high coverage. However, in all the scenarios explored, either the girls-only 2-dose or the GNV 2-dose strategies had the highest probability of being cost-effective at 0.5 GDP threshold and above.

Table 4 shows the cost-effectiveness frontier of the expected values of costs and QALYs for the base-case and alternative distributions, and the three coverage scenarios. For the base-case distributions, in all coverage scenarios, the no vaccine and both 1-dose strategies are dominated (costs and QALY components are higher and lower, respectively, than the dominant strategy). The girls-only 2-dose is cost-saving and the GNV 2-dose ICERs are all well below 1xGDP WTP threshold for all coverage scenarios. For the alternative distribution, in the low coverage scenario, the dominance structure is the same compared to the base-case

**Table 4. Cost-effectiveness frontier for expected values of costs and QALYs for the base-case and alternative distributions for each coverage scenario.**

|  | Strategy | Base-case Distributions | | | Alternative Distributions | | |
| --- | --- | --- | --- | --- | --- | --- | --- |
|  |  | Cost | QALYs | ICER (ΔC/ΔQ) | Cost | QALYs | ICER (ΔC/ΔQ) |
| **Low Coverage** | No Vaccine | 57.53829 | 23.93433 | Dominated | 57.53829 | 23.93433 | Dominated |
|  | Girls-only 1-Dose | 56.65939 | 23.94532 | Dominated | 54.14560 | 23.95498 | Dominated |
|  | GNV 1-Dose | 58.01421 | 23.94728 | Dominated | 55.11395 | 23.95841 | Dominated |
|  | Girls-only 2-Dose | 55.41127 | 23.95287 | - | 53.99855 | 23.95832 | - |
|  | GNV 2-Dose | 57.22536 | 23.95595 | 589 | 55.64290 | 23.96205 | 441 |
| **Medium Coverage** | No Vaccine | 57.53829 | 23.93433 | Dominated | 57.53829 | 23.93433 | Dominated |
|  | Girls-only 1-Dose | 55.98209 | 23.95689 | Dominated | 50.70203 | 23.97571 | - |
|  | GNV 1-Dose | 59.24473 | 23.95991 | Dominated | 53.14027 | 23.98013 | Dominated |
|  | Girls-only 2-Dose | 52.54066 | 23.98017 | - | 50.99127 | 23.98508 | 31 |
|  | GNV 2-Dose | 58.12686 | 23.98455 | 1,277 | 56.31299 | 23.98917 | 1,299 |
| **High Coverage** | No Vaccine | 57.53829 | 23.93433 | Dominated | 57.53829 | 23.93433 | Dominated |
|  | Girls-only 1-Dose | 55.85785 | 23.96029 | Dominated | 50.40513 | 23.98151 | - |
|  | GNV 1-Dose | 59.79840 | 23.96344 | Dominated | 54.05266 | 23.98574 | Dominated |
|  | Girls-only 2-Dose | 51.62934 | 23.99260 | - | 51.28738 | 23.99404 | 70 |
|  | GNV 2-Dose | 59.70711 | 23.99595 | 2,411 | 59.41892 | 23.99719 | 2,583 |

distributions (no vaccine and 1-dose programs are still dominated), and the ICER for GNV 2-dose is lower for the alternative distribution, which is because most of the vaccinated population in the low coverage scenario received only 1-dose. However, for medium and high coverage scenarios the dominance structure changes with the alternative distributions resulting in girls-only 1-dose program showing cost-saving, GNV 1-dose remains dominated (weakly), the girls-only 2-dose has a very low ICER ($31/QALY and $70/QALY), and the GNV 2-dose ICERs are still well below the 1xGDP WTP threshold.

## Cost-effectiveness-sensitivity analyses

The sensitivity analyses indicate that cost-effectiveness results are most sensitive to variations in coverage, dose price, and discount rate, and not very sensitive to the treatment cost or health state utility variations (see more details Appendix G in S1 Appendix). The cost-effectiveness acceptability for the base-case distribution for different per dose cost indicate that both 1-dose strategies have very low probability of being cost-effective in any of the coverage or dose cost variations, and that as either dose price or coverage increases, so does the range of WTP values for which girls-only 2-dose programs have a high probability of being cost-effective.

The cost-effectiveness acceptability for different discounting indicates that both 1-dose strategies have very low probability of being cost-effective in any of the coverage or discount variations, and that as discounting and coverage increase, so does the WTP range for which girls-only 2-dose is most likely cost-effective. The cost-effectiveness acceptability for different treatment costs indicates that, within each coverage scenario, there is no significant difference in the cost-effectiveness acceptability curves under variations of treatment cost, and that for all coverage and treatment cost scenarios, 1-dose strategies are dominated, girls-only 2-dose strategy is cost-saving, and the ICER of GNV 2-dose strategy is not very sensitive within coverage scenarios. Finally, the cost-effectiveness acceptability for different health state utilities values is similar to that of the treatment cost variations.

We also performed all the above sensitivity analyses using the alternative distributions (see Appendix G 2.2 in S1 Appendix). These results indicate that, in general, the probability of 1-dose programs being cost-effective is higher in terms of acceptability curves, particularly for low WTP thresholds and for scenarios with high coverage, high dose price or high discount rate. However, the 2-dose programs are still more likely to be cost-effective at reasonable WTP thresholds.

## Discussion

This study used a DTM of HPV vaccination to assess the impact on cancer cases avoided and the cost-effectiveness of four different vaccination strategies, which included 2-dose or 1-dose regimens for girls-only or gender-neutral vaccination strategies. The model was calibrated to the Indonesia population, as an example, to illustrate the impact these strategies can have in a typical LMIC setting. To factor in possible variations in implementation and completion of vaccination programs, the analysis explored low, medium, and high coverage scenarios.

The main objective of the study was to assess the public health impact and cost-effectiveness of different dosing strategies for HPV vaccination programs with a 9-valent vaccine, while rigorously accounting for the current uncertainty in the effectiveness of a single dose of vaccine. The effectiveness of a single dose of vaccine was represented by the degree of protection and waning parameters in our model. The uncertainty in these parameters was characterized by a joint probability distribution estimated using a novel Bayesian approach and the 36-month vaccine efficacy data from the KEN SHE trial, the first available RCT specifically studying single dose efficacy. In addition, we estimated vaccine model properties using KEN SHE data

combined with the 10-year follow-up effectiveness data from the India IARC study. This approach enabled us to infer the probability of success of adopting a 1-dose program relative to 2-dose programs.

Previous versions of the model were improved herein, such that waning (i.e., durability of vaccine protection) was modelled as a Gamma distribution (instead of exponential) to better reflect the existing data on single-dose efficacy, and to be more aligned with the waning profile used in other single-dose models [11, 40, 41, 45]. However, to our knowledge, this is the first HPV vaccine modeling study that formally inferred, using a Bayesian framework, single-dose vaccine properties based on clinical trial data, whereas other models typically relied on more *ad hoc* assumptions about these properties, albeit informed by clinical trial or observational data.

Our results indicate that cervical cancers account for the majority of cancers avoided for any of the vaccination scenarios explored. Factors leading to this result, in contrast to many high-income settings, is the lack of existing national immunization programs (NIPs), lack of robust historical cervical screening programs, disparities in access to healthcare services, and the potential underreporting of non-cervical cancers in low- and middle-income settings. The results indicate that, in all scenarios explored, the 2-dose regime strategies lead to larger number of cancer cases avoided, on average, compared to the corresponding 1-dose strategies. The differences in median cases averted between the 1- and 2-dose programs become larger with higher coverage and completion rates, although the uncertainty around the 1-dose estimates also increases with coverage and completion rates. When focusing on all HPV-related cancer cases, the results also show that the GNV programs always avoid more cancer cases than the girls-only programs (when comparing same dose programs). Moreover, when focusing on non-cervical cancers, the 2-dose GNV program avoid significantly more cancer cases than the 2-dose girls-only program. Lastly, when using the alternative distributions (longer protection), as expected, the median number of cases avoided by 1-dose increases and moves closer to that of the 2-dose, but also that the GNV 1-dose program tends to avoid more male cancer cases, on average, than the girls-only 2-dose program, and that the range of variability in health outcomes under the alternative distributions is reduced but remains significant. In summary, the 2-dose strategies lead to larger health outcomes benefits compared to the 1-dose strategies, on average, but with a large amount of variability around the 1-dose health outcomes, which is driven by the uncertainty in the evidence around the long-term protection of a single dose.

With respect to cost-effectiveness, these results indicate that the 2-dose girls-only strategy can be cost-saving and the 2-dose GNV strategy is cost-effective for a wide range of scenarios, and that 1-dose GNV strategy is generally cost-effective relative to 1-dose girls-only strategy. Thus, on the basis of cost-effectiveness, GNV programs should be considered a preferred program/strategy by policy makers in LMIC settings, with budget constraints not-withstanding.

Under the alternative distribution scenario, for medium and high coverage, the 1-dose girls-only strategy can be cost-saving, and thus is the economically preferred strategy when the WTP threshold is close to 0 (less than about $70/QALY). However, in most settings, the WTP threshold is often higher and as high as 1xGDP, in which case the economically optimal strategy is the one with an ICER below the chosen WTP threshold. Our results indicate that both the girls-only 2-dose and GNV 2-dose strategies have ICERs well below the 1xGDP WTP threshold and thus provide more population health benefits for relatively little additional cost.

While this work is focused on the potential health and long-term economic risk of a single-dose program given the uncertainty in 1-dose effectiveness, in the real world, policy makers often consider many other factors that could influence this decision, which are not included in this analysis, such as budgetary constraints, logistical barriers, equity considerations,

negotiated per-dose price (which may be higher for 1-dose) among other considerations, that will influence the ultimate choice of vaccination program and its health and economic success.

In our main analysis we did not focus on cervical cancer elimination since the analysis would entail examining the introduction of screening programs that do not currently exist in Indonesia as in many other middle- and low-income countries. However, we do present some elimination results in Appendix G.3 in S1 Appendix. Elimination at 4 cases per 100,00 women-years is likely to occur, in the base-case high coverage scenario for 2-dose GNV at a median of 71 to 74 years and not likely to occur within 100 years for the 1- or 2-dose programs in the base-case for all other coverage scenarios and strategies. This result holds for the alternative distribution as well. A recent UNFPA report [63] noted that Indonesia could eliminate cervical cancer by 2074 if there is a concerted effort in implementing three elimination pillars of HPV vaccination, screening and pre-cancer treatment, and cervical cancer treatment.

As noted in the Introduction section and detailed in Appendix A in S1 Appendix, previous models have considered 1-dose vaccination programs in the LMIC context. Generally, they have characterized 1-dose performance making point value assumptions about the duration and degree of protection (or "take"). Using the same point value vaccine parameterization and other assumptions in our model should result in similar conclusions; for example, that 1-dose is cost-effective compared to 2-dose if the vaccine durability is >20 years (with perfect protection) or the degree of protection or vaccine "take" is over 80% (with permanent protection) and high enough dose price. However, the probability of 1-dose vaccine parameters having these values is low in either of our fits to the KEN SHE and India-IARC data (see Table 3). This implies a very low probability of achieving the associated health and cost-effectiveness outcomes estimated by those models. Beyond some very generic statements, however, it is difficult to compare our results directly with the results of existing models, given that we use different coverage and dose price assumptions, as well as different disease- and economic-related model parameter values, all specific to the Indonesia setting (which is not the case for other existing models). In addition, most other health-economic modeling for LMIC settings consider only cervical cancer diseases, while our model covers all the major HPV-related diseases and thus provide a more comprehensive comparative assessment of the health and economic impacts of the different vaccination strategies.

This study is subject to limitations. First, the model assumes that the degree of protection reduction factor is the same for all disease areas, both sexes, HPV types, and endpoints. Second, we estimated our joint probability distributions for 1-dose vaccine properties based on the arms of the KEN SHE study, which reported only cervical persistent infection for either pooled HPV 16/18 types or pooled HPV 16/18/31/33/45/52/58 types. These two assumptions likely overestimate the single-dose effectiveness against other disease endpoints considering that, in multi-dose studies that include other endpoints, the efficacy is generally observed to be lower than the cervical persistent infection endpoint [6, 52, 64]. Third, we assumed that the per-dose price for a single-dose program would be the same as a multi-dose program. This is unlikely to be true outside of the GAVI context as actual NIP dose pricing is the result of negotiation that may result in lower volume discounting for fewer doses as well as other factors that affect dose pricing negotiations. Fourth, we did not consider the introduction of cervical screening and treatment programs, either as a means for monitoring the 1-dose program, or for accelerating the time to elimination. However, there are substantial barriers to implementing these programs in LMICs [65–67]. Fifth, we assumed that coverage (the % of individuals with 1 or more doses) were the same in the 1-dose and 2-dose strategies; all else equal, the 1st dose compliance rates may be marginally larger in a single-dose program as compared to a 2-dose program, although there is limited empirical evidence suggesting that initiation rates will be larger in settings with a vaccination program with fewer recommended vaccine doses.

For example, the change in HPV vaccine recommendation from 3 doses to 2 doses in 2016 in the USA, and 2 doses to 1 dose in 2022 in the United Kingdom (UK), did not seem to have a significant effect on the national trends of initiation rates in the subsequent years [68–71]. Moreover, the introduction of a 1-dose HPV program in 2023 in Australia was associated with a decrease in vaccine initiation that year [72, 73]. Given that the difference in initiation rates between a 1- and 2-dose program depends on multiple factors and may be highly variable across countries and settings, until further evidence is available, any direct comparison between "1-dose with high coverage" and "2-dose with low coverage" scenarios would be based on unrealistic expectations and could result in misleading conclusions. We also did not model the idea of replacing the 2-dose girls-only program with a 1-dose program plus cervical screening and treatment because, as noted above, the introduction of a new screening and treatment program has substantial barriers that may take several years to implement, and will likely require substantially more funding to implement and maintain than the funds that may be saved in vaccine costs in a 1-dose instead of 2-dose HPV vaccine program. Finally, as with most statistical analyses, the uncertainty of the model parameters captures both the uncertainty in the efficacy data as well as "structural" uncertainty due to the model choice. We mitigated some of this uncertainty by exploring multiple vaccine model structural assumptions and used Bayesian Information Criterion (BIC) to support our selected model structure. As a result, our estimates for duration of protection are, effectively, lower-bounds on the true duration–longer term trial data may extend our duration estimates but is less likely to shorten it. Other general model limitations have been discussed in previous publications [48, 57].

## Conclusions

These analyses indicate that, in a LMIC, a 2-dose vaccination program in adolescents, either GNV or girls-only, are more effective than their 1-dose counterparts at reducing the number of HPV-related cancers. Furthermore, the 2-dose programs are much more likely to be cost-effective than 1-dose programs for a wide range of scenarios and WTP thresholds. While 1-dose strategies can be cost-saving in high coverage scenarios, 2-dose strategies may still be preferred due to the ability to provide greater health benefits while maintaining cost-effectiveness. Due to the rapidly evolving evidence on this topic, health economic models need to be updated as additional trial results become available.

## Supporting information

**S1 Appendix.**
(PDF)

## Acknowledgments

Assistance with manuscript review was provided by Didik Setiawan, Alain Luxembourg, Alfred Saah, Elamin Elbasha, and Ya-Ting Chen. Medical writing assistance was provided by Molly Gingrich and Abigail Zion, employees of Analysis Group.

## Author Contributions

**Conceptualization:** Vincent Daniels, Kunal Saxena, Oscar Patterson-Lomba, Andres Gomez-Lievano, Jarir At Thobari, Nancy Durand, Evan Myers.

**Data curation:** Vincent Daniels, Kunal Saxena.

**Formal analysis:** Vincent Daniels, Kunal Saxena, Oscar Patterson-Lomba, Andres Gomez-Lievano.

**Investigation:** Vincent Daniels, Kunal Saxena, Oscar Patterson-Lomba, Andres Gomez-Lievano.

**Writing – original draft:** Vincent Daniels, Kunal Saxena, Oscar Patterson-Lomba, Andres Gomez-Lievano.

**Writing – review & editing:** Vincent Daniels, Kunal Saxena, Oscar Patterson-Lomba, Andres Gomez-Lievano, Jarir At Thobari, Nancy Durand, Evan Myers.

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
