## [Decision Letter · Decision Letter 0]

16 Jul 2024

PONE-D-24-06862Modeling the health and economic implications of adopting a 1-dose 9-valent human papillomavirus vaccination program in adolescents in low/middle-income countries: an analysis of IndonesiaPLOS ONE

Dear Dr. Daniels,

Thank you for submitting your manuscript to PLOS ONE. After careful consideration, we feel that it has merit but does not fully meet PLOS ONE’s publication criteria as it currently stands. Therefore, we invite you to submit a revised version of the manuscript that addresses the points raised during the review process.

We have received valuable comments that could make your paper stronger. Look carefully on the reviewer comments and provide the necessary responses and revised versions.

We look forward to receiving your revised manuscript.

Kind regards,

Jonah Musa, MBBS, MSCI,PhD

Academic Editor

PLOS ONE

Journal Requirements:

"I have read the journal's policy and the authors of this manuscript have the following competing interests:

Vince Daniels and Kunal Saxena are employees of Merck Sharp & Dohme LLC, a subsidiary of Merck & Co., Inc., Rahway, NJ, USA and shareholders in Merck & Co., Inc., Rahway, NJ, USA.

Oscar Patterson-Lomba and Andres Gomez-Lievano are employees of Analysis Group, Inc., a consulting company that has provided paid consulting services to Merck & Co., Inc, which funded the development and conduct of this study and manuscript.

Jarir At Thobari, Nancy Durand, and Evan Myers have received consultancy fees from Merck & Co., Inc."

Reviewers' comments:

Reviewer's Responses to Questions

**Comments to the Author**

1. Is the manuscript technically sound, and do the data support the conclusions?

Reviewer #1: Yes

Reviewer #2: Yes

2. Has the statistical analysis been performed appropriately and rigorously? 

Reviewer #1: Yes

Reviewer #2: Yes

3. Have the authors made all data underlying the findings in their manuscript fully available?

Reviewer #1: Yes

Reviewer #2: Yes

4. Is the manuscript presented in an intelligible fashion and written in standard English?

Reviewer #1: Yes

Reviewer #2: Yes

5. Review Comments to the Author

Reviewer #1: Thank you for the opportunity to review this manuscript. The approach and overall quality of the manuscript is good. However, I have the following comments on the structure, findings and conclusions of this work:

1. First, from the outset, I would like to make a general statement. The whole discussion around one-done HPV vaccination strategies is based on logistical and sustainability considerations, especially with experience from low and middle-income countries that have recently introduced the vaccine. Without adequate coverage, even a three-dose vaccination dose program would not achieve the desired outcomes. In the 90:70:90 framework, while what "fully-immunized" may change as more evidence becomes available, the 90% coverage threshold may not.

2. Cost of vaccines: the authors state that their vaccine costs inputs were ranging from USD 10-60, yet the current estimates of the gardasil-9 vaccine for instance is around USD 287. Why not use the most realistic vaccine prices, for a country that would be seeking guidance from a paper like this on what policy to adopt? Unless I have missed it, I haven't seen any other different vaccine cost in the model inputs section of the methods. What was the vaccine cost assumption based on?

3. I think the authors need to have considered additional scenarios for the comparative analyses. For instance, comparing low coverage with a two-dose strategy vs high coverage with a one-dose strategy. A cursory look at figure 1 actually shows that a 1-dose strategy with high coverage actually outperforms a two-dose strategy with low coverage in cervical cancer cases verted. This is the true dilemma many LMICs face. If this work is to be seen as balanced and impartial, the authors should evaluate and discuss this in detail.

4. In lines 402-412, the authors clearly state that while the two-dose strategy is cost-saving in all coverage scenarios, in high dose scenario, the 1-dose strategy becomes cost-saving but the 2-dose one has very low ICER. Since this represents the nearest picture to the true situation in low and middle-income countries of different contexts, this should be discussed and made part of the conclusion.

5. One argument for adoption of 1-dose strategies is that financial resources saved can be invested in other public health programs like screening and treatment, which would have impact on incidence and mortality of cervical cancer earlier than vaccination. What would be the budget impact of adopting a one-dose strategy as opposed to a two-dose strategy, especially for no-Gavi supported countries or those transitioning out of support? Even though the model adapted in this study was not able to do this additional scenarios, it would add value to cover this in the discussion section, since that's what decision-makers really need to know.

6. The decision to adopt the KENSHE results into the model is a good one. Are the authors willing to update their model as more data from the trail (follow-up is continuing) become available?

7. Because of number 4 above, I find the conclusion rather shallow or even misleading.

Reviewer #2: Thank you for the chance to review this interesting paper. I believe these results will be beneficial to the field moving forward. I have a few comments that I believe will make the paper stronger.

1. Clearer description of why the KEN-SHE data were used in conjunction with data from IARC India earlier in the methods section. This is mentioned, but could be clearer

2. While results of the study indicate cost-effectiveness for a 2-dose vaccination program, the discussion would benefit from some comments about how realistic this would be for a single-payer system, and barriers that may impact real-world cost-effectiveness of such a vaccination program

3. The primary limitation of the model is that it assumes that the degree of protection is equal for all disease areas, both sexes, HPV types, and endpoints. This is a major limitation that requires some additional comments. Is there literature that provides context about how 1-dose vs. 2-dose regimens might have varying impact on these characteristics? What guardrails would be valuable for implementing this type of program in similar LMICs given this limitation? How can this limitation be better addressed in future research?

4. The authors did not model the introduction of cervical cancer screening programs, which is a reasonable decision as it is out of scope for the topic being investigated. As such, it is my opinion that the statement about LMICs being “unable to introduce robust and organized screening programs”, should be removed as it minimizes the sociocultural, political, and health system barriers to organized screening in LMICs which also influence vaccination uptake, and discounts the value of the grassroots efforts aimed at increasing awareness about cervical cancer and the importance of screening.

6. PLOS authors have the option to publish the peer review history of their article (what does this mean?). If published, this will include your full peer review and any attached files.

Reviewer #1: No

Reviewer #2: No

---

## [Author Response · Author response to Decision Letter 0]

28 Aug 2024

Please see the attached "Response to Reviewers" file for a detailed response to all reviewer comments.

---

## [Editor Report · Decision Letter 1]

3 Sep 2024

Modeling the health and economic implications of adopting a 1-dose 9-valent human papillomavirus vaccination program in adolescents in low/middle-income countries: an analysis of Indonesia

PONE-D-24-06862R1

Dear Dr. Daniels,

We’re pleased to inform you that your manuscript has been judged scientifically suitable for publication and will be formally accepted for publication once it meets all outstanding technical requirements.

Kind regards,

Jonah Musa, MBBS, MSCI,PhD

Academic Editor

PLOS ONE

Additional Editor Comments (optional):

Thank you for the detailed responses to the comments raised by the reviewers, and for incorporating the required revisions in the revised version. I am pleased with your overall efforts and responsiveness to the reviewers.
---

## [Editor Report · Acceptance letter]

9 Sep 2024

PONE-D-24-06862R1 

PLOS ONE

Dear Dr. Daniels, 

I'm pleased to inform you that your manuscript has been deemed suitable for publication in PLOS ONE. Congratulations! Your manuscript is now being handed over to our production team.

Kind regards, 

on behalf of

Dr. Jonah Musa 

Academic Editor

PLOS ONE